# Noradrenaline and acetylcholine shape functional connectivity organization of NREM substages: An empirical and simulation study

Fernando Lehue 🆔[1,2], Carlos Coronel-Oliveros[3,4], Vicente Medel[3], Thomas Liebe[5,6], Jörn Kaufmann[7], Sebastián Orellana[1,2], Diego Becerra[1,2], Enzo Tagliazucchi[3,8], Patricio Orio 🆔[1,9]*

**1** Centro Interdisciplinario de Neurociencia de Valparaíso, Valparaíso, Chile, **2** Programa de Doctorado en Ciencias, mención Biofísica y Biología Computacional, Universidad de Valparaíso, Valparaíso, Chile, **3** Latin American Brain Health Institute (BrainLat), Universidad Adolfo Ibáñez, Santiago, Chile, **4** Global Brain Health Institute (GBHI), Trinity College Dublin, Dublin, Ireland, **5** Department of Psychiatry and Psychotherapy, University of Jena, Jena, Germany, **6** Department of Dermatology and Venerology, University Hospital Magdeburg, Magdeburg, Germany, **7** Department of Neurology, University of Magdeburg, Magdeburg, Germany, **8** Buenos Aires Physics Institute and Physics Department, University of Buenos Aires, Buenos Aires, Argentina, **9** Instituto de Neurociencia, Facultad de Ciencias, Universidad de Valparaíso, Valparaíso, Chile

* patricio.orio@uv.cl

**Data availability statement:** fMRI data has been previously published in Tagliazucchi et al., 2012 (10.1016/j.neuroimage.2012.06.036) and

## Abstract

Sleep onset is characterized by a departure from arousal, and can be separated into well-differentiated stages: NREM (which encompasses three substages: N1, N2 and N3) and REM (Rapid Eye Movement). Awake brain dynamics are maintained by various wake-promoting mechanisms, particularly the neuromodulators Acetylcholine (ACh) and Noradrenaline (NA), whose levels naturally decrease during the transition to sleep. The combined influence of these neurotransmitters on brain connectivity during sleep remains unclear, as previous models have examined them mostly in isolation or only in deep sleep. In this study, we analyze fMRI data obtained from healthy individuals and employ a whole-brain model to investigate how changes in brain neurochemistry during NREM sleep, specifically involving ACh and NA, affect the Functional Connectivity (FC) of the brain. FC analysis reveals distinct connectivity changes: a decrease in Locus Coeruleus (LC) connectivity with the cortex during N2 and N3, and a decrease in Basal Forebrain (BF) connectivity with the cortex during N3. Additionally, compared to Wakefulness (W), there is a transition to a more integrated state in N1 and a more segregated state in N3. Using a Wilson-Cowan whole-brain model, informed by an empirical connectome and a heterogeneous receptivity map of neuromodulators, we explored possible mechanisms underlying these dynamics. We fit the model adjusting the coupling and input-output slope of the whole-brain model to account for ACh and NA, respectively, and show that region-specific neurotransmitter effect is key to explain their effects on FC. This work enhances our understanding of neurotransmitters' roles in modulating sleep

is availabe at https://zenodo.org/records/16755776 Structural connectome (kindly provided by Dr. Gustavo Deco (Deco et al., 2018, 10.1016/j.cub.2018.07.083)) and Python codes to reproduce analysis and Figures are available at https://github.com/vandal-uv/NREMmodFC.

**Funding:** F.L. is supported by Beca de Doctorado Nacional N° 21220708 (Agencia Nacional de Investigación y Desarrollo ANID, Chile). V.M. is supported by FONDECYT Exploración 13240170 and FONDECYT de Iniciación 11251578 (ANID, Chile). T.L. is supported by a German Research Foundation Grant (D.F.G.) Projektnummer 41466077, 526259959. J.K. received no specific funding for this work. S.O. is supported by Beca de Doctorado Nacional N°21241572 (ANID, Chile). D.B is supported by Beca de Doctorado Nacional N°21210914 (ANID, Chile). E.T is supported by FONDECYT Exploración 13240170 and FONDECYT Regular 1220995 (ANID, Chile). P.O. is supported by FONDECYT Regular 1241469 (ANID, Chile) Fondecyt Exploración 13240170, and the Advanced Center for Electrical and Electronic Engineering (AFB240002 ANID, Chile).The funders had no role in study design, data collection and analysis, decision to publish, or preparation of the manuscript.

**Competing interests:** The authors have declared that no competing interests exist.

stages and their significant contribution to brain state transitions between different states of consciousness, both in health and disease.

## Author summary

Falling asleep involves a gradual shift away from wakefulness and into distinct stages of sleep: NREM (with stages N1, N2, and N3) and REM (Rapid Eye Movement). Awake brain activity is promoted by neurotransmitters like acetylcholine (ACh) and noradrenaline (NA), which decrease during sleep. However, how these chemicals shape brain connectivity during sleep isn't fully understood.

In this study, we used brain scans (fMRI) from healthy individuals to explore how ACh and NA influence brain connectivity during NREM sleep. We found that Basal Forebrain and Locus Coeruleus –main sources of ACh and NA, respectively– reduce their connections with the rest of the brain in deeper sleep stages. We also saw that the brain areas become more connected in light sleep (N1) and less during deep sleep (N3).

To better understand these patterns, we used a computer model of the whole brain, combining real brain structure data with maps showing where ACh and NA have the most influence. Our results suggest that these neurotransmitters play a key role in how the brain shifts between wakefulness and different sleep stages—insights that may help us better understand sleep-related conditions and consciousness itself.

## Introduction

Sleep is a natural state of reduced consciousness, where the brain transits between different functional states naturally driven by neuromodulatory systems [1]. Transitions between different stages of sleep constitute a naturalistic way to understand the neurochemical mechanisms behind states of consciousness. The patterns of static and dynamic BOLD-derived FC (as measured by fMRI) change during the different stages of sleep; there are spectral [2], dynamic connectivity [3], and static connectivity changes [4,5] that shape brain architecture during sleep. Characterizing these neural dynamics and understanding how they emerge can provide insights into how consciousness arises from neural processes, how it is lost, and can even suggest ways to recover it [6,7].

The brain shows marked differences in connectivity between wakefulness (W), NREM, and REM phases [8]. More specifically in NREM sleep, the differences in static FC can be used to accurately classify sleep stages [9]. In [10], the authors report a decrease in thalamocortical connectivity during N1, which is partially restored in N2 and N3, suggesting a thalamic origin of sleep-specific neural signatures, like spindles [11] that orchestrate these changes. The differences in FC between sleep stages have also been characterized in the Integration-Segregation axis [12,13], which are two principles that capture how the brain organizes different sources of information processing [14,15]. It is generally assumed that integration and segregation work in a balance that facilitates flexibility in cognitive demands, and previous studies have connected the effects of Acetylcholine (ACh) and Noradrenaline (NA) on modulating this balance, with ACh and NA promoting segregated and integrated states, respectively [16,17]. How these changes arise from brain-wide neurotransmitter modulation has not been fully explored.

Neuromodulators mediate the changes associated with the transition from wakefulness to sleep [1]. Neuromodulators differ from classical neurotransmitters in that they are typically expressed by unique clusters of neurons, they project diffusely throughout the nervous system, and they modulate postsynaptic neurons in a way that alters their responses to traditional neurotransmitters (such as gamma-aminobutyric acid GABA, and glutamate) [1]. The Ascending Activating System (AAS) encompasses the main nuclei that release these neuromodulators [18], coordinating their response to different natural contexts, and sustaining arousal and the sleep cycle. The main neuromodulators of wakefulness that are reduced during NREM sleep are ACh and NA, while GABA has been associated with sleep initiation mechanisms [1].

ACh is a neurotransmitter that acts as a neuromodulator in the central nervous system (CNS), playing an important role in arousal, attention, memory, and motivation [19,20]. The two main cholinergic nuclei of the brain are the Pedunculopontine nucleus (PPN), and the Basal Forebrain (BF), each with distinct functions and projections, but which together work to shape brain function. Tonic discharge of ACh from the BF neurons is highest during wakefulness and REM sleep, and lowest during NREM sleep [21]. It promotes a desynchronization of neuronal field potential activity, as measured in EEG [21]. ACh acts through two classes of receptors: muscarinic (mAChRs, metabotropic) and nicotinic (nAChRs, ionotropic). Although having many effects, a relevant consequence of the activation of pre- and post-synaptic ACh receptors is an enhanced excitation and reduced cortico-cortical interactions [19,20,22].

Noradrenaline (NA) is an organic chemical of the catecholamine family, whose function is to promote rapid responses to environmental changes and is the main modulator of arousal [23]. According to the Glutamate Amplifies Noradrenergic Effects (GANE) model [24], arousal-induced NA released from the LC biases perception and memory in favor of salient, high-priority representations at the expense of lower-priority ones, through the different adrenergic receptors [25]. This effect can be locally summarized as an increase in the slope of the input-output function of the excitatory population, having a sharper "all or none" response [26]. This makes the neuronal population more sensible to suprathreshold incoming stimuli, and less responsive to subthreshold ones. LC neurons discharge of NA is state-dependent and lower during sleep: LC neurons display the highest discharge rates in W, lower during NREM, and are virtually silent during REM sleep [27]. However, recent works in rodents have shown that NA levels fluctuate dynamically, reaching even higher levels in thalamus in NREM sleep than W [28].

Evidence has shown that changes in brain connectivity in sleep can arise due to changes in neuromodulation by ACh and NA [1], and these changes can be characterized by differences in the balance of integration and segregation of the brain network. Previous empirical and modeling studies have investigated the effect of ACh in the transition from wakefulness to deep sleep [3,29], where ACh and NA modulation has been proposed to sustain integration and segregation dynamics [16,17].

In this work, we aim to connect sleep modulation by neurotransmitters and FC differences in sleep, proposing a comprehensive model of NREM substages modulation. From empirical analysis, we hypothesize a differential neuromodulation profile and a fluctuation of integration and segregation across NREM stages, and a testable mechanistic connection between these two observations. We tested these hypotheses using a Wilson-Cowan model of brain dynamics, where we varied parameters that represent the effect of ACh and NA. Further, we hypothesized that the effects of ACh and NA in mediating the transitions from wakefulness to sleep depend on the specific connectivity of neuromodulatory nuclei with the brain [30,31]. To test this hypothesis, we incorporated BF and LC projections to the

brain as priors, and compared empirical and simulated BOLD-derived FC matrices as well as integration/segregation profiles as objectives of fit.

## Results

We used fMRI data previously obtained in [9]. Recordings were obtained from 71 individuals, simultaneously with polysomnography measures from which sleep stages were labeled in each fMRI volume. As is common during a normal night's sleep, there are regular jumps between epochs of different sleep stages. For our purposes, all fMRI volumes of the same individual in the same sleep stage were concatenated, obtaining different time series for W, N1, N2 and N3 stages. Individuals with time spent in all four stages were selected for analysis (N=15 individuals). BOLD data voxels were averaged over larger cortical and sub-cortical Regions of Interest (ROIs), determined by the AAL90 brain parcellation (see Materials and Methods). In addition to this, we applied a mask to obtain the BOLD time courses of BF [32] and LC [33]. AAL90-parcellated, BF and LC timeseries are publicly available in https://zenodo.org/records/16755776.

### BF and LC functional connectivity changes from wakefulness to sleep

We calculated the $90 \times 90$ FC matrices from the extracted BOLD time series, defined as the Pearson correlation coefficient between all pairs of brain areas. As previously reported [34], we observe an overall variation in the FC distribution between different areas of the brain as a function of sleep depth. We extracted the BOLD activity of LC and BF, and compared the FC between these nuclei and the cortex (LC-FC and BF-FC, respectively) between W and NREM stages. During N1, BF-FC and LC-FC remain around the same levels as in W (effect sizes of $D = -0.17$ for BF and $D = 0.22$ for LC, neither significant); there is a stronger decrease in LC-FC during N2 ($D = -0.19$ not significant for BF and $D = -0.92$ significant for LC), and a joint decrease of BF-FC and LC-FC during N3 ($D = -1.06$ significant for BF and $D = -1.9$ significant for LC) (see Fig 1A, significance at Benjamini-Hochberg corrected $p < 0.05$, we report the repeated measures correlation value for comparing multiple groups [35]).

Compared to W, Cohen's Ds between node strengths are $D = 1.09$ in N1, $D = 0.05$ in N2 and $D = -1.9$ in N3. When analyzing the spatial distribution of these effects, for W we see a strong BF-FC to frontal regions, and mild BF-FC to occipital regions, that decrease to mild and low in N3, respectively. In the case of the LC connectivities with the cortex, we see a more homogeneous initial correlation structure that decreases in N2 and N3 in an approximately uniform manner (see brain connectivity plots in Fig 1B, generated using Brainnet Viewer [36] in MATLAB). For reference, see MNI-aligned masks of BF and LC in Fig 1C.

When comparing nodal strength (sum of incoming FC to each node) against W, we find the highest increase in light sleep N1, which decreases in N2 and N3, with a minimum in this latter stage. All these fluctuations are significant (T-test, $p < 0.05$, Benjamini-Hochberg correction for multiple comparisons) (see Fig 1D).

These results suggest that even if FC profiles may not fully capture the transitions between sleep stages, they might be affected by changes in the neuromodulatory influences of the cholinergic and noradrenergic systems. We further test this idea using computational modeling with mechanistic assumptions about the role of neuromodulation.

### Testing BF and LC local modulation of NREM stages in-silico

A whole-brain computational model was implemented. The brain is separated in 90 areas according to the AAL90 parcellation, and the activity of each area was simulated using a

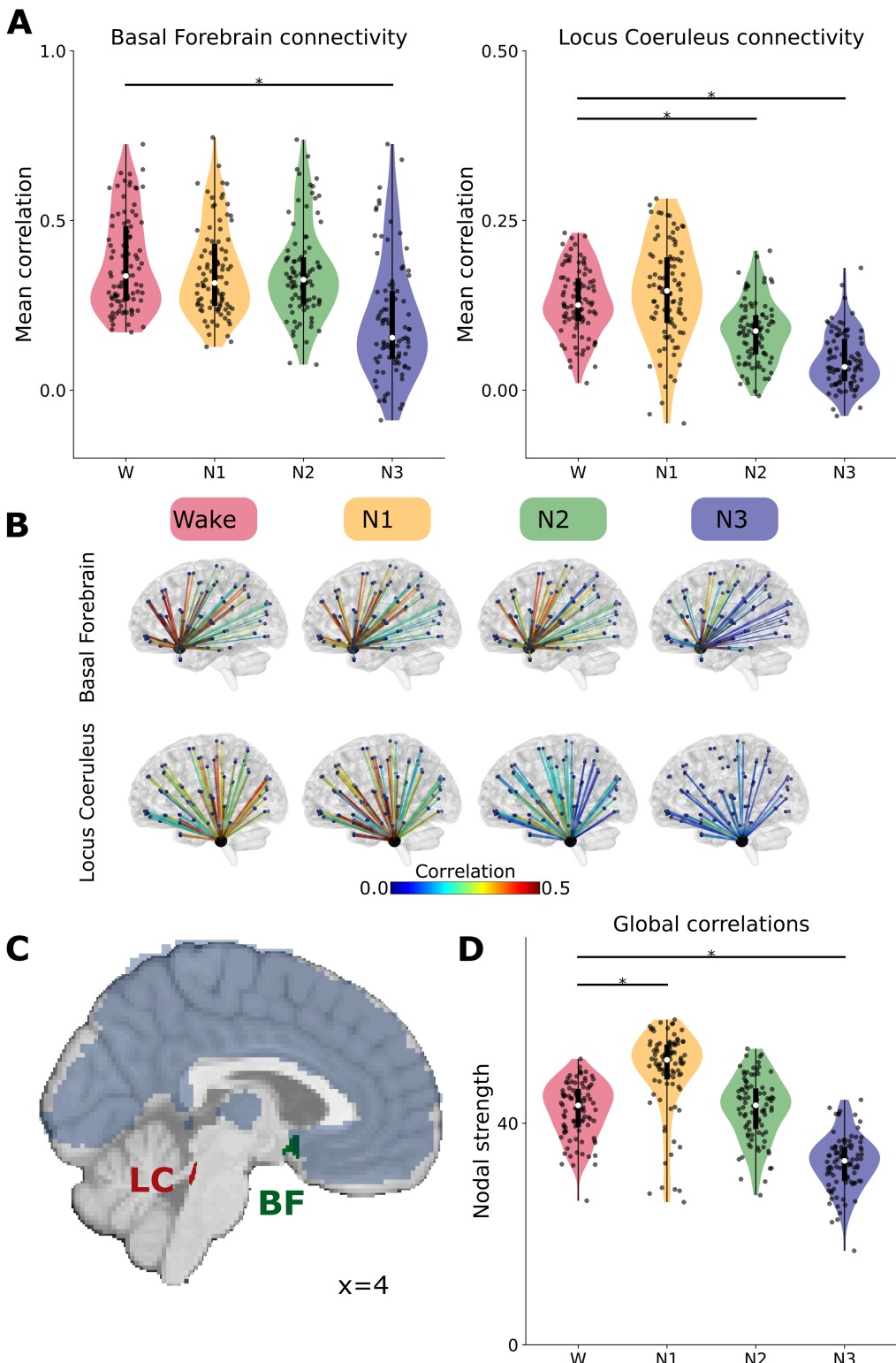

**Fig 1. Brain-wide connectivity changes in W and NREM stages. (A)** FC between brain areas and BF, and FC between brain areas and LC, across sleep stages. Each point is the value for one brain area, averaged across individuals (multiple comparisons). Horizontal black lines represent significance (T-test, $p < 0.0001$). **(B)** Glass brain plots, displaying the spatial distribution of FC with BF and LC, correlation value displayed in color. **(C)** LC and BF position in the brain. AAL90 areas displayed in light blue (not individually labeled). **(D)** Functional nodal strength (sum of incoming FC values) across sleep stages.

Wilson-Cowan oscillatory model with homeostatic inhibitory plasticity [37]. Excitatory populations were connected to each other according to an empirical human brain connectome [31]. According to previous modeling approaches, ACh tone was modeled as a decrease of the global coupling parameter $G$ of the model [17,22,38,39], and NA tone as an increase in the slope parameter $\sigma$ of the input-output function of each brain area [26]. Instantaneous excitatory activity of each node was fed into a Balloon-Windkessel model of hemodynamic response of brain oxygenation to neuronal activity [40], to obtain a simulated BOLD signal (Fig 2, see Materials and Methods for more details).

We first fitted the model to the empirical averaged FC matrix of individuals in the awake (W) state. The fit was obtained with a 2D sweep of $G$ and $\sigma$ parameters, minimizing the euccorrelation metric between empirical and simulated FC (we developed the euccorelation metric as a trade off between Euclidean distance and Pearson correlation, see Materials and Methods). The $G$ and $\sigma$ values that best fitted the W state were used as reference for the rest of fitting, and therefore the optimal parameters for N1, N2 and N3 FC fit were expressed as variations from those of W ($\delta_G$ and $\delta_\sigma$)

**Homogeneous modulation.** As a first exploration, we considered a scenario of homogeneous neuromodulation, in which the effect of ACh and NA were assumed to be equal in

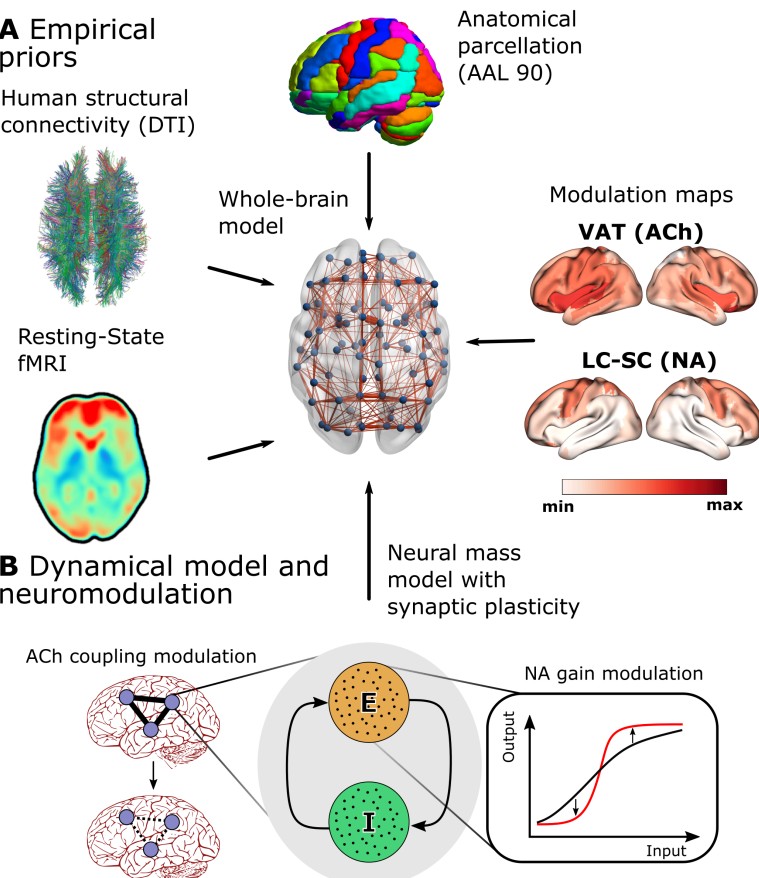

**Fig 2. Modeling outline. (A)** Empirical priors included to construct our whole-brain model. **(B)** Nodal dynamics scheme, exemplifying how ACh and NA hypotheses are included. ACh is taken as a modulator of global cerebral coupling, while NA modulates the input-output slope of each node.

all brain regions. In other words, the parameters $G$ and $\sigma$ were uniformly changed across all nodes.

Compared to W optimal, a higher $G$ parameter is required for fitting N1, that then is lower (to the exact same as in W) in N2, and a much lower $G$ value for N3 ($\delta_G = 0.04$ in N1, $\delta_G = 0$ in N2, and $\delta_G = -0.04$ in N3, see left in Fig 3A and 3B). For $\sigma$, we do not observe variations until N3, that needs a higher value ($\delta_\sigma = 0$ in N1 and N2, and $\delta_\sigma = 0.04$ in N3, right in Fig 3A and 3B). These are the "naïve" variations that our model suggests, before considering the distinct effect of NA and ACh over different brain areas, and could be interpreted as a decrease in ACh in N1, a null change of ACh and NA in N2, and a simultaneous increase in ACh and decrease in NA in N3 (see homogeneous simulated FC matrices in the center column of Fig 4).

**Heterogeneous modulation.** Then we moved on to test the hypothesis that the region-specific modulation of ACh and NA are important for reproducing the empirical data, making the modulation of $G$ and $\sigma$ heterogeneous according to a spatial Modulation Map of each neurotransmitter.

We used the empirical structural projections from Locus Coeruleus to the cortex (LC SC, [41]) for regionally leveraging the effects of NA, and the Vesicular Acetylcholine Transporter (VAT [42,43]) for ACh (see Brain maps in Methods). We repeated the sweep of $\delta_G$ and $\delta_\sigma$, but with a heterogeneous regional prior given by these respective maps. Regional values of the $G$ and $\sigma$ parameters are given by equation (5). Running 50 seeds, we compared the fit (euccorrelation value) in the optimal parameter combination with homogeneous versus heterogeneous modulation, using Cohen's D [44].

Heterogeneous variation of parameters sets a new W optimal from which to compare the other stages (optimal W: $\delta_G = 0$ and $\delta_\sigma = -0.02$). From this point, the transition from W to N1, N2 and N3 requires a reduction of ACh modulation ($\delta_G = 0.18, 0.2, 0.02$ for N1, N2 and N3, respectively), in parallel to a decrease of NA tone ($\delta_\sigma = -0.02, -0.04, -0.12$ for N1, N2 and N3, respectively). These values can be interpreted as a decrease in ACh in N1, N2 and N3 (although with an unexpected high variation in N1, see Discussion), and a decrease of NA starting in N2, that is deeper in N3 (see homogeneous simulated FC matrices in the rightmost column of Fig 4).

In order to test whether the specific regional distribution of the maps improves the fit, we performed the same procedure with a randomized version of the maps (See Map Shuffling in Materials and Methods). In all sleep stages, the best fit –i.e., lower distances to empirical FC – was obtained in the heterogeneous map case, and the shuffled map produced the worst fits in N1 and N3 (Fig 3C). As measured by Cohen's D and its range interpretation (see Methods), the greatest improvement given by map heterogeneity was in N1 (huge), N3 (huge), W (large) and N2 (medium) (see values in Fig 3C).

## Integration and Segregation changes during sleep

Given the proposed relation between ACh and NA modulation with integration and segregation [16], and given our claim that ACh and NA modulation could specifically characterize each NREM sleep stage, we sought to explore whether empirical integration and segregation change through sleep stages, and how ACh and NA neuromodulatory systems are associated with FC network properties.

For quantifying integration and segregation, we calculated the integration and segregation nodal components of empirical and simulated FC matrices, using the Hierarchical Modular Analysis framework [45], which is a partition method based on eigenmodes of human brain

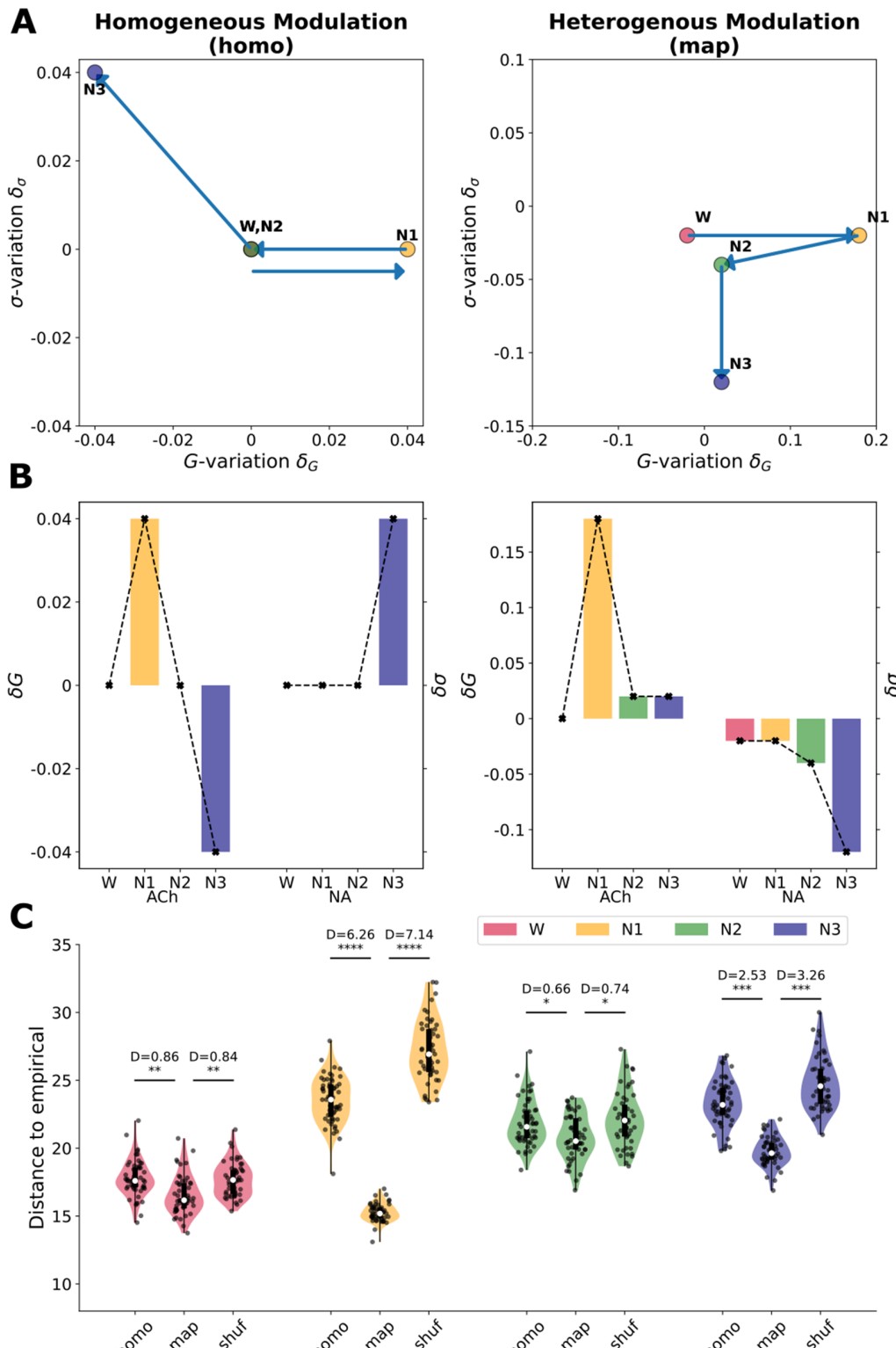

**Fig 3. Optimal parameters and goodness of fit across stages and modalities.** (A) Optimal parameter variation ($\delta_G$ and $\delta_\sigma$), where the euccorrelation metric between empirical and simulated FC matrices is minimized, displayed for all stages in the homogeneous and heterogeneous simulation cases. Arrows represent the direction of change from one state to the next (note that for homogeneous modulation W and N2 are in the same spot). (B) Optimal parameters for the homogeneous and heterogeneous simulation cases, displayed side by side for each stage, for ease of view. Notice the apparent "flipping" of the y axis for the coupling parameter, which arises because we are modeling a decrease in ACh as an increase in G. (C) Violin plots comparing the euccorrelation metric between the simulated and empirical

FC matrices, in the optimal, across 50 seeds, for all stages (* for $p < 10^{-3}$, ** for $p < 10^{-4}$, *** for $p < 10^{-22}$, **** for $p < 10^{-50}$, see Cohen's D interpretation in Methods).

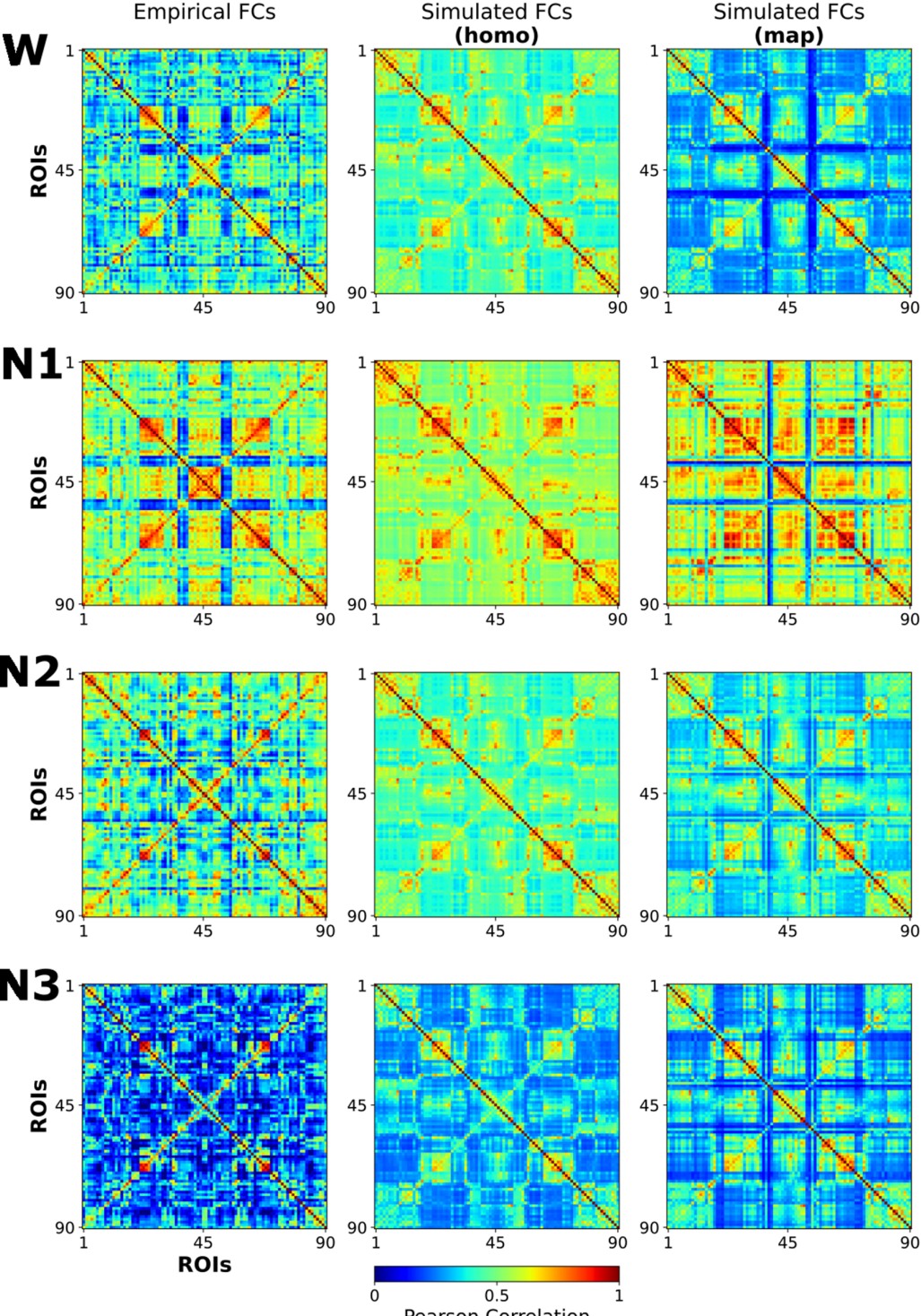

**Fig 4. Empirical and optimal simulated matrices across stages and modalities.** Empirical and simulated optimal whole-brain FC matrices for the homogeneous and heterogeneous simulation case, displayed side by side, for all stages (rows).

functional networks, that measures the global integration and segregation of the network and their local component for each ROI (see Methods).

When analyzing the global trend, we notice a transition towards an integrated FC from W to N1, which returns to lower values in N2, and then reverses to a FC dominated by segregation in N3 (Fig 5A).

We took the BF-FC and LC-FC of each area, and compared them with their respective nodal integration and segregation components (local weighted contribution of each region to the whole-network value of integration and segregation, see Methods) in the W state, when ACh and NA modulation are highest. We found a positive correlation between the segregation component of brain areas and the BF-FC profile ($\rho = 0.4784$ Pearson's correlation coefficient, $p < 0.0001$), and also a positive correlation between their integration component and the LC-FC ($\rho = 0.6243$, $p < 0.0001$) (Fig 5B). This suggests that, in the awake brain, areas with a higher FC to LC tend to have a higher integration component, and that areas with a higher functional interaction with BF tend to have a higher segregation component. For completeness, note that BF-FC correlates negatively with integration and LC-FC correlates negatively with segregation, as expected (see small sub-panels within Fig 5B).

**In-silico integration and segregation.** To further test our hypothesis of NA and ACh neuromodulation as a mechanistic explanation for the observed FC changes during sleep, we tested the ability of the model to replicate these empirical results, measuring the similarity of simulated and empirical profiles of integration and segregation. We calculated the Pearson's correlation coefficient between the optimal model's integration and segregation profile, across 50 simulation seeds, for the homogeneous, heterogeneous with modulatory map, and random heterogeneous cases. In particular, we quantified how much heterogeneity improves or worsens the correlation with empirical data (Cohen's D effect size calculated over Pearson's correlation coefficients). We found that simulated integration and segregation in the heterogeneous map case improved the fit to empirical integration and segregation in all stages (Fig 5C and Table 1; see S1 Fig for the full distribution of integration and segregation profiles in empirical and simulated data).

## Discussion

### Summary

In this work, we analyzed empirical FC of individuals in NREM sleep stages, and fitted FC using a whole-brain model that includes ACh and NA neuromodulatory influences. From the analysis of the fluctuations of empirical FCs, our results suggest a decrease in NA modulation during N2, and a joint decrease of ACh and NA modulation in N3. At the same time, they do not show changes in LC and BF FC between W and N1. The in-silico testing of our empirically-derived hypotheses of NREM modulation then suggests a role of ACh in the transition from W to N1, and a role of NA in the transition from N1 to N2 and N3 (see parameters in Fig 3 and FC matrices in Fig 4). When leveraging the influence of NA and ACh using Modulation Maps, we better reproduce empirical FC across all brain states, suggesting that the transitions from W to different NREM stages are mediated by regional-specific NA and ACh modulation. We also found that, compared to W, the organization of the brain network shows an increase in integration in N1, that in N2 shifts back to values close to those of W, and a shift towards a network dominated by segregation in N3. Furthermore, we found evidence that ACh and NA modulate the integration-segregation of the brain in an antagonistic fashion (Fig 5B). Finally, the model is capable of describing the empirical integration-segregation distribution across sleep stages, and even more when informed with ACh and NA modulation maps (Fig 5C).

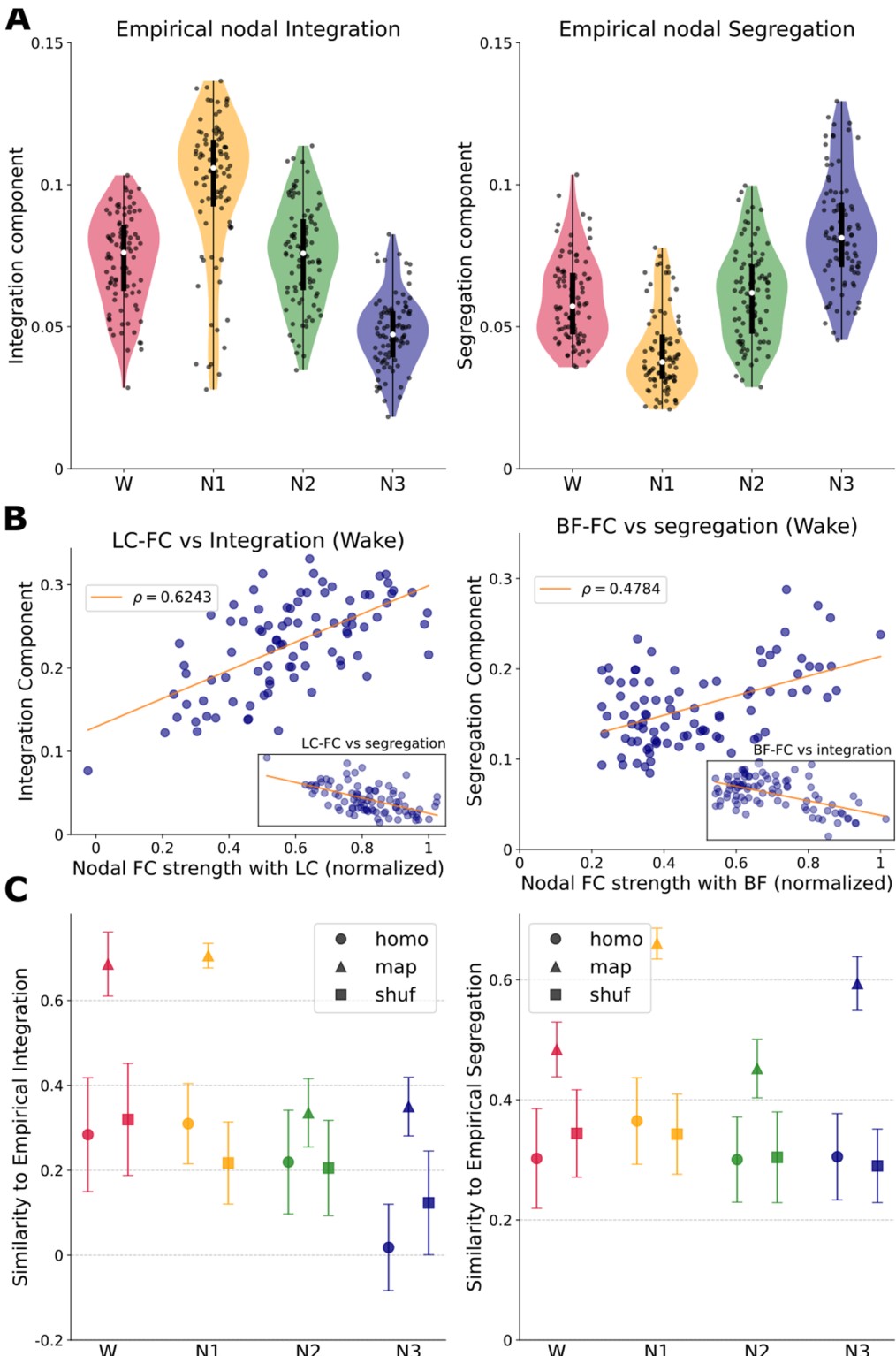

**Fig 5. LC and BF effects on the integration/segregation profiles in W and NREM. (A)** Empirical nodal integration and segregation components across stages. Each dot is the value for one brain area, averaged across individuals. **(B)** Empirical nodal integration component versus nodal FC connectivity with the LC, and nodal segregation component versus nodal FC connectivity with the BF, in W. $\rho$ is the Pearson's correlation coefficient between the variables. LC-FC vs segregation ($\rho = -0.64$) and BF-FC vs segregation ($\rho = -0.54$) are shown in the small panels within. **(C)** Correlation between simulated and empirical integration (segregation) components, for all stages and modalities (homo, map and

shuffle). Each point is the correlation between the simulated integration (segregation) component of one seed with the integration (segregation) component of the average empirical FC of the corresponding stage. A higher value implies a better fit to the empirical integration (segregation) profile.

**Table 1. Cohen's D effect size between the empirical-simulated similarity measure (Pearson's coefficient), measuring how much better or worse one modality of simulation is over another (for example, map-homo D = 3.88 for N1 Integration means that simulations with map have consistently higher correlation with empirical Integration distribution than homogeneous simulations, with a Cohen's D value of 3.88).**

| | Integration | | Segregation | |
|---|---|---|---|---|
| | map-homo | map-shuffle | map-homo | map-shuffle |
| W | 3.66 | 3.37 | 2.69 | 2.28 |
| N1 | 5.61 | 6.77 | 5.4 | 6.22 |
| N2 | 1.11 | 1.32 | 2.47 | 2.3 |
| N3 | 3.78 | 2.26 | 4.77 | 5.61 |

## Perspectives on neuromodulation

There is extensive literature on the implication of ACh and NA in the modulation of arousal, and sleep onset [1,46,47]. However, other than their different effects in general NREM vs REM sleep, few studies have found a finer dissociation of ACh and NA in modulating NREM substages. For example, in [21] the authors address the role of ACh in early sleep, through its influence on thalamocortical interactions in wakefulness and general NREM sleep. In [29], the authors investigate the role of ACh in the emergence of slow waves in a computational brain model, but without including N1 and N2 stages nor NA modulation in their analyses. In another study in rodents, the optogenetic stimulation of basal forebrain ACh neurons during early NREM sleep promotes a transition to wakefulness, while late stimulation promotes transition towards REM sleep [48]. Although we do not analyze REM sleep here, the fact that our results suggest a lower ACh modulation in NREM stages than in W is in agreement with these studies.

Likewise, during NREM sleep, phasic 0.5-1 Hz burst firing of LC NA-neurons has been reported in rodents, coupled with slow waves and anti-correlated with spindles [49,50]. Those neurons project to the prefrontal cortex and thalamus, where a diminished activity promotes the transition to REM sleep only when it occurs late in rodent NREM sleep [51]. Considering N1-N3 are characterized by higher and higher stimulation thresholds for awakening [52], it is natural to hypothesize the deepest decrease of NA in the (exclusively human) N3 stage, which is what our results suggest.

However, it is important to note that rodent studies have revealed region-specific nuances in NA dynamics. For instance [28] showed that tonic NA levels in the sensory thalamus were actually higher during NREM sleep compared to quiet wakefulness in mice, highlighting that NA is not reduced everywhere in the brain during sleep. These findings support the idea that neuromodulatory dynamics are different in cortical and subcortical structures, and suggest caution when generalizing global trends from regional to global data or from rodents to humans. As the majority of the nodes in our model are cortical, we do not explicitly distinguish between cortical and subcortical nodes. Moreover, our framework for map inclusion only allows for all nodes to be modulated in the same direction (higher or lower). Thus, we must interpret our results primarily as reflecting large-scale cortical effects of neuromodulatory gradients.

Although many sleep-facilitating drugs work through GABAergic agonism [53], different factors affect vigilance and sleep onset, involving many neurotransmitters acting

simultaneously and interacting in complicated ways [1,54,55]. Serendipity plays a big role in drug discovery [56]. Cholinergic agonists (like Donepezil, commonly prescribed for the treatment of Alzheimer's cognitive symptoms) usually cause insomnia-like side effects, and at the same time an overall change of sleep architecture toward a predominance of REM sleep [1,57]. An example of the intricacies of sleep modulation is that of clonidine and mirtazapine; while the former is an agonist of $\alpha_2$-adrenergic receptors, and the latter an antagonist, these two medications have sleep-promoting side effects. We believe that although we do not focus on specific receptor effects, our results are interpretable as the general effect of tonic levels of NA and ACh, and our approach to disentangle their modulation of NREM sleep stages can be taken as substrate, or even extended, for future empirical or in-silico analyses that delve deeper into the neurochemistry of sleep, the way in which FC is organized differently in NREM substages, and how this is reflected in integration and segregation measures of brain connectivity.

Although the great majority of cholinergic innervation of the cortex comes from the BF, only a small proportion of BF neurons are cholinergic (10-20% in rat), and they are concentrated in the Meynert subnucleus portion of the BF [58], whose cortical innervation has been shown to closely correlate with Vesicular Acetylcholine Transporter (VAT) [43]. On the other hand, LC is more homogeneously populated by noradrenergic neurons [59], so we hypothesize that a structural projection map coming from LC is representative of the NA modulation exerted by this nucleus. This is the explanation for the apparent discrepancy between using a map of the VAT distribution for ACh [43], but structural projections of LC for NA [41].

## Integration and segregation in sleep

Previous research [12,60] found decreased integration and increased modularity during late NREM sleep (stages S2-4). Simultaneous EEG and fMRI recordings show that mutual information between brain activity patterns progressively decreases from wakefulness to N3 [61]. Likewise, our findings show that N2 is more similar to wakefulness in terms of segregation/integration balance. We found that in N3, integration is lower and segregation higher, which we connect to NA modulation, and that adds to the growing body of literature suggesting that brain activity in N3 is more local than in the other stages [60,61]. In our work, ACh seems to promote an increase of integration in early sleep, while decrease in NA induces an increase of segregation later, in line with their effects in wake dynamics already described in the literature in rodents [16,27,51].

Overall, in sleep research, different measures (from graph theory, information theory, and spectral analysis) might be capturing similar neuromodulatory mechanisms that we here try to jointly encompass. To arrive at more comprehensive conclusions, common datasets using the same criteria to differentiate NREM sleep stages must be analyzed with similar measures and techniques.

## Noradrenaline and acetylcholine fluctuations during NREM may help to explain fluctuations in consciousness

Sleep is a reversible model of altered consciousness, distinct from coma, epilepsy, or anesthesia due to its cyclical and naturally induced nature. Brain activity and responsiveness change gradually across sleep stages (N1 to N3), highlighting that consciousness is not binary but multidimensional. Our work builds on previous literature about how wake-promoting neurotransmitters affect vigilance in specific ways, and how their interaction can originate these different states of connectivity and stability. Dreams (i.e., distorted consciousness during sleep) have been reported not only during REM sleep, but also in NREM

sleep [62–67], albeit with less vivid imagery reported in general [68]. Dreaming (and its recall) also correlates with reduced slow wave activity and increased fast spindles, specially in posterior parietal and central cortical areas [66]. As slow waves tend to be anti-correlated with acetylcholine [69], and thalamic noradrenaline is anticorrelated with sleep spindles[28], dreams might be expected when acetylcholine is high (for NREM standards) and noradrenaline is low. This is coherent with current mechanism hypotheses such as apical drive due to cholinergic modulation [70]. Also, lower noradrenaline levels might correlate with inability or difficulty to remember dreams [71]. Understanding fine-grained neuromodulation dynamics might shed light on how and when subjective experience is modified or lost during sleep, contributing to the ongoing discussion regarding dreamless sleep [72–74]. Also, more detailed simulation including noradrenaline fluctuations across NREM stages both in cortex and thalamus would allow us to make more fine-grained predictions in order to explain how, and when, can dreams occur within NREM sleep.

Finally, studying how neuromodulation can mediate these transient changes in consciousness might help us in understanding how structural and/or neuromodulatory permanent disturbances can be related to impaired unconsciousness, like Disorders of Consciousness [7,75]. We expect our empirical and in-silico approach could be applied to assess hypotheses of the emergence of other states of consciousness to gain insights into their origin and modulation.

### Strengths and limitations

It is important to acknowledge the limitations of our study. While computational models offer a powerful tool for studying complex biological systems [76], they inherently rely on simplifications and assumptions that may not fully capture the intricacies of real-world physiology. Our model, for instance, makes simplifications for the functioning of neurotransmitter systems, and does not take into account the influence of other neurotransmitters, such as GABA, and other external factors, such as regulators of the circadian rhythms or environmental stimuli.

We made strong assumptions in the way we model NA and ACh's effects in our model. These have been used in previous works [17,77], and are based on biological evidence [22,26]. For NA, our assumptions build on the GANE framework [24], which, despite its simplicity, has been referred in recent studies to interpret both human and rodent data [78,79]. For ACh, we implemented its effects as a reduction in global coupling, inspired by its proposed role in enhancing local processing and reducing long-range correlations [22]. Although we believe that the results concerning the specific effect of the spatial distribution of NA and ACh modulation are solid, future studies could aim to incorporate more realistic effects of these neurotransmitters to provide a more comprehensive understanding of sleep regulation.

Results for N1 are conflicting, and we believe that could be explained by the existence of other phenomena that our model is not capturing, like thalamic-driven short-time modulation of cortical dynamics [80], the appearance of up and down states of the local population of neurons in deep sleep (that has been connected to NA modulation dynamics) [50], and the role of other neuromodulators whose levels also change in sleep [1,46]). The short duration of N1 compared with the other stages makes its exploration more difficult, and while we know that ACh levels are higher in wakefulness and REM, and lower in N2 and N3, N1 is seldom mentioned in studies that classify data in substages (e.g. [81]). In [9], the highest staging error from the fMRI FC matrices was the misclassification of W as connectivity N1, suggesting that N1 shares many similarities with W, which could by itself be explained by the fact that eyes closed wakefulness also displays general higher connectivity than eyes open [82], or that it is

systematically misclassified compared to W, N2 and N3. These factors could lead to distortions in the expected optimal parameters. It is worth noticing that also when the ACh parameter of the model is fixed on its N1 optimal value, a decrease on NA modulation is needed to transition to N2 and N3 (see sweeps for all modalities and stages in S1 Fig), which is in accordance to NA's assumed role in sleep modulation [50,51,83,84]. We believe that in future studies, more suitable metrics and fitting approaches can disentangle the interplay between NA, ACh, GABA, serotonin, and other modulators' effects in sleep in a way that also includes electrophysiology. All in all, we are proposing ACh and NA modulation mechanisms that explain transitions between sleep stages, but that may not be enough to explain the transition to light sleep N1.

## Conclusion

In this work, we conclude that heterogeneous neuromodulation that uses an anatomically informed prior produces a better fit of the FC observed in each NREM sleep stage, above both a homogeneous and a randomized heterogeneous neuromodulation. This result confirms our hypothesis that anatomically informed priors of NA and ACh influences explain the changes in FC across sleep stages.

Our study contributes to the growing body of literature on neurotransmitter modulation of NREM sleep stages by proposing specific roles of ACh and NA in sleep architecture. Using computational modeling, we have demonstrated how alterations in ACh and NA levels can impact the brain-wide connectivity profile of NREM stages, also in terms of integration-segregation balance, and offering potential avenues for therapeutic intervention in sleep disorders. Further research in this area holds promise for advancing our understanding of sleep physiology and developing targeted treatments to improve sleep health.

## Materials and methods

### EEG-fMRI acquisition and preprocessing

The dataset comprises EEG, EMG and fMRI recordings acquired from 71 participants [9]. All subjects were scanned during the evening and instructed to close their eyes and lie still and relaxed.

A cap EEG device with 30 channels (sampling rate 5 kHz, low pass filter 250 Hz) was used during fMRI acquisition (1505 volumes of $T^*$-weighted echo planar images, TR/TE = 2,080 ms/30 ms, matrix $64 \times 64$, voxel size $3 \times 3 \times 2$ mm$^3$, distance factor 50%, field of view [FOV] 192 mm$^2$) at 3 T (Siemens Trio) with a polysomnographic setting, consisting on Electromyography (EMG) on the chin and tibia, Electrocardiogram (ECG), bipolar Electrooculography (EOG), and pulse oximetry. MRI artifact correction was carried out based on the average artifact substraction (AAS) method [85]. Using Statistical Parametric Mapping (SPM8) EPI data were realigned, normalized (MNI space) and spatially smoothed (Gaussian kernel, 8 mm$^3$ full width at half maximum). Sleep staging was performed by an expert according to the AASM criteria [86].

A more detailed description of demographics, scanning parameters, and experimental conditions are provided in [5].

### Brain parcellation and structural connectivity estimation

Participants brains were parcellated into 90 cortical Regions of Interest (ROIs), according to the cerebral labels of the Automated Anatomical Labeling Atlas (AAL90 [87]), in line with

previous whole-brain modelling work [31]. AAL90 subdivides the entire brain into 76 cortical and 14 subcortical regions, 45 per hemisphere. The structural connectivity between brain areas was obtained using diffusion Magnetic Resonance Imaging, and resources and the methods are available in [31].

Previous to all experiments carried out in this work, we performed an optimization of homotopic connectivities of the structural connectivity, i.e., the connection of each area with its corresponding counterpart in the other hemisphere, following the general scheme of [88]. This is because DTI has been reported to underestimate interhemispheric connectivity, which is also reflected as a mismatch between empirical and simulated FC matrices. The procedure consists in fitting an optimal coupling parameter of a whole-brain model to the empirical W FC, and then iteratively simulating FC matrices in epochs $t \in \{1, 2, ..., 60\}$, updating entries of the original SC matrix according to the following rule:

$$SC^t_{i,j} \mapsto SC^t_{i,j} + \epsilon(FC_{i,j;\text{empirical}} - FC^t_{i,j;\text{simulated}}),$$

where $i,j$ are restricted to homotopic connections. Notice that the simulated FC matrix is the mean of 20 seeds, for canceling out stochastic noise. The Hopf dynamical model was used for this procedure, for its simplicity and speed (see [88] for more details).

## FC estimation

For ACh and NA modulation nuclei, we used previously obtained masks for extracting the fMRI time courses of BF and LC [32,33]. Furthermore, due to the proximity between the pedunculopontine nucleus (PPN) and the LC, and the considerably larger size of the former compared to the latter, we regressed out the PPN time courses from the LC ones across all subjects using a PPN-specific MNI mask [33], which has been suggested to control for possible confounding effects of PPN on LC activations [55,89]. After this, we computed the FC connectivity vectors between BF and LC with AAL90 ROIs across subjects and brain stages (W, N1, N2 and N3 sleep) as the Pearson correlation between all areas. We used these values for subsequent analysis.

## Brain maps

The effects of NA and ACh at the whole-brain level are complex, and are achieved by the contribution of many of their receptors (such as $\alpha 1$ and $\alpha 2$ for NA, and Muscarinic and Nicotinic for ACh). For simplicity, we hypothesized that the structural projections of the respective nuclei should parsimoniously encompass the effects of these neurotransmitters.

In the case of NA we used the structural projections from the LC to each AAL ROI as a Modulation Map (Locus Coeruleus Structural Connectivity LC-SC). This data was extracted in [41], and the authors kindly provided it in the AAL90 parcellation. For this, The AAL (ROIs) were transferred from MNI space to the individual subject space using ANTs (https://picsl.upenn.edu/software/ants/), and the connectivities were recalculated for each subject from the voxel level in the AAL parcellation, and averaged across subjects.

The maps were normalized so that their mean was 1, in this way their change in parameter, in the mean, was equal to their modulation by $\delta$ explained further in this article. A bit different was the case of the LC structural map, which presented values 20 times higher in the thalamus compared to the other areas. For being able to interpret the effects of the map in all non-thalamic areas, LC connectivity values for the thalamus were de-escalated to the maximum of the other areas before normalized.

For ACh, we used the Vesicular Acetylcholine Transporter distribution (VAT), obtained using PET [42,43], which has been shown to have a close correlation with cholinergic innervations coming from the cholinergic Meynert subnucleus of the BF [90].

For obtaining the FC distribution of ROIs with the BF and LC, the BOLD time series of these two areas were extracted for each awake fMRI volume, and then correlated with all AAL90 ROIs, obtaining a 90-valued vector of correlations for each subject. After this, these vectors were averaged across subjects, obtaining a representative map of the FC distribution of the BF and LC with the rest of the brain.

## Whole-brain model

The Wilson-Cowan model is an ODE-based neural-mass model for the activity of an excitatory and inhibitory population of neurons. As a model, it has been widely used for simulating brain activity, due to its relative simplicity but high effectivity and flexibility in modeling brain dynamics [17,37,91]. In our whole-brain model setting, each brain area of our parcellation is modeled by a Wilson-Cowan oscillator, that are later connected through an empirically obtained connectome matrix [31]. The activity of the Excitatory $E$ and Inhibitory $I$ subpopulations of one brain area are defined using differential equations, which, for the $i$–th node are:

$$\tau^E \frac{\mathrm{d}E_i}{\mathrm{d}t} = -E_i + (1 - r^E E_i)S\left(a^{EE}E_i - a_i^{IE}I_i + \sum_{j=1}^{N} C_{ji}E_j + P + D\varepsilon\right) \tag{1}$$

$$\tau^I \frac{\mathrm{d}I_i}{\mathrm{d}t} = -I_i + (1 - r^I I_i)S\left(a^{EI}E_i\right) \tag{2}$$

$$\tau^{ip} \frac{\mathrm{d}a_i^{IE}}{\mathrm{d}t} = I_i(E_i - \rho^E) \tag{3}$$

Where $r^E = r^I = 0.5$ are self-response constants $\tau^E = 0.01$, $\tau^I = 0.02$ are excitatory and inhibitory time constants, and $a^{EE} = 3.5$ and $a^{EI} = 3.75$ are self-excitatory and excitatory-to-inhibitory, intra-node coupling constants, respectively. The matrix $C_{ij}$ represents the excitatory-to-excitatory long-range coupling values (the corresponding connections between the respective areas $i,j$ in our parcellation) [31]. $P = 0.4$ is the external input to the excitatory population, taken to be equal for each area. $\varepsilon$ is zero-centered white noise, and $D = 0.002$ is the corresponding noise scaling factor.

Equation (3) corresponds to the implementation of a homeostatic inhibitory plasticity mechanism, in which the inhibitory-to-excitatory intra-node connection $a_i^{IE}$ evolves so that the corresponding excitatory variable $E_i$ oscillates in the vicinity of a set value of $\rho^E = 0.18$. Inhibitory plasticity has a time constant of $\tau^{ip}$. This plasticity mechanism is based on biophysics and grants the model a more robust and stable response to fluctuations and changes in parameters [37].

The integration of the incoming stimulus (noise and input or from other nodes) for a node $j$ is made through its evaluation in a sigmoid-like activation function $S_{E,I}$, whose defining equation is:

$$S_{E,I}(x) = \frac{1}{1 + e^{-(x-\mu)/\sigma_{E,I}}} \tag{4}$$

where $\mu = 1$, $\sigma = 4$ are position and slope constants, respectively. It is important to notice that when we consider variations of the $\sigma_E$ parameter, it changes its value only for the excitatory population, and $\sigma_I = 4$ remains constant.

Time units are seconds. Equations were integrated using the Euler method scheme for simulating differential equations with a $dt$ = 0.0001. White noise was sampled at each time step. A transient period of 400 seconds was simulated first with a fast inhibitory plasticity ($\tau^{ip}$ = 0.05), and then discarded to reach stable dynamics. After this, 600 seconds were considered for analysis, simulated with $\tau^{ip}$ = 2. Excitatory activity $E_i$ was fed into a hemodynamic model [92] to obtain a simulated BOLD response.

## fMRI BOLD signals simulation

Once instantaneous excitatory activity for each brain area $E_i(t)$ was obtained, we simulated BOLD-like signals using the Balloon-Windkessel model of hemodynamic response [92]. An increase in excitatory activity starts a vasodilatory response $s_i$, which in turn triggers blood inflow $f_i$, and changes in blood volume $v_i$ and deoxyhemoglobin content $q_i$. The equations governing these responses are

$$\frac{ds_i(t)}{dt} = E_i(t) - \frac{s_i(t)}{\tau_s} - \frac{f_i(t)-1}{\tau_f}$$

$$\frac{df_i(t)}{dt} = s_i(t)$$

$$\tau_v \frac{dv_i(t)}{dt} = f_i(t) - v_i(t)^{1/\kappa}$$

$$\tau_q \frac{dq_i(t)}{dt} = \frac{f_i(t)\left(1 - (1-E_0)^{1/f_i(t)}\right)}{E_0} - \frac{q_i(t)v_i(t)^{1/\kappa}}{v_i(t)}.$$

Time constants of signal decay, blood inflow, blood volume, and deoxyhemoglobin content are $\tau_s$ = 0.65, $\tau_f$ = 0.41, $\tau_v = \tau_q$ = 0.98, respectively. The resistance of veins and arteries to blood flow is represented by a stiffness constant $\kappa$ = 0.32, and the resting-state oxygen extraction rate is $E_0$ = 0.4. The BOLD response $B_i(t)$ is a non-linear function of $q_i(t)$ and $v_i(t)$, given by

$$B_i(t) = V_0 \left[ k_1(1 - q_i(t)) + k_2\left(1 - \frac{q_i(t)}{v_i(t)}\right) + k_3(1 - v_i(t)) \right]$$

where $V_0$ = 0.04 represents the ratio of venous (deoxygenated) blood to all blood in resting-state, and $k_1$ = 2.77, $k_2$ = 0.2, $k_3$ = 0.5 are kinetic constants.

This system of differential equations was solved using the Euler method, with an integration step $dt$ = 0.001 seconds. These BOLD simulated signals were then filtered in the frequency range [0.01,0.1] with a 2nd order Bessel filter, and downsampled to $TR$ = 2, as empirical data. We used these signals to build the FC matrices, taking the pairwise correlations between all AAL90 brain areas.

## ACh and NA maps modulation

The effects of ACh and NA were introduced in the model as modulating the coupling $G$ and input-output slope $\sigma$, respectively. For an ACh (NA) map distribution $ACh_i$ ($NA_i$), we gave the coupling (slope) parameter a local value $G_i$ ($\sigma_i$) for each brain area $i$, expressed as a variation from the homogeneous value $G$ ($\sigma$), weighted by a delta $\delta_G$ ($\delta_\sigma$) parameter:

$$G_i = G + \delta_G \cdot ACh_i \tag{5}$$

$$\sigma_i = \sigma + \delta_\sigma \cdot NA_i.$$

The $\delta_G$ parameter was swept taking 50 equidistant parameter values in the range $[-0.5, 0.5]$. The $\delta_\sigma$ parameter took 50 equidistant parameter values in the range $[-1, 1]$. 50 runs with different seeds were simulated for each change of parameters.

## Map shuffling

To obtain a surrogate distribution of values for a given map (90 values, 45 per hemisphere), we randomly shuffled its values. The procedure was performed symmetrically in each hemisphere, so its correlation with structural connectivity (which is highly homotopically symmetric itself) was preserved. This shuffling procedure ensured that the average value of the maps per hemisphere was conserved.

## Fit distance: The euccorrelation semi-metric

We assessed the Goodness of fit of our model as its capacity to reproduce empirical FC matrices. Once generated, we compared empirical and simulated FCs using a metric based on Euclidean distance and Pearson Correlation, which we call the *euccorrelation*, being able to capture both the overall connectivity pattern, and overall connectivity strength between simulated and empirical FC matrices (being a trade off between Euclidean distance and Pearson correlation, as [93]), while also having the advantage of being invariant under changing the order of data ROIs. Since FC matrices are symmetric, we only took the values under the diagonal and compared them as flattened vectors. The *euccorrelation* metric is defined as:

$$euccorr(v1, v2) = \frac{euc(v1, v2)}{|\rho(v1, v2)|}, \tag{6}$$

where $euc(v1, v2)$ is the Euclidean distance between our flattened-FC-matrix vectors $v1$ and $v2$, and $\rho(v1, v2)$ is the Pearson Correlation between their values. As for measuring similarity, this measure is a lower fit when Euclidean distance is lower and correlation is higher, so it is to be minimized to obtain a better fit to empirical data.

## Hierarchical modular analysis

To identify functional modules in both empirical and simulated data, and quantify their integration and segregation components, we used the Hierarchical Modular Analysis (HMA) method following [45,94]. In short, the method applies an SVD decomposition of the FC matrix to find its eigenvectors and eigenvalues. The regions whose corresponding entries in the eigenvector have the same signs are assumed to have joint activity (cooperation) and put in the same module, and another one for the negative signs. The first functional level (first eigenvector) has one module that encompasses all brain areas, the second level divides the brain in two modules according to the signs of the entries of the second eigenvector, the third in four modules, and so on. During this partition process, the number of modules in each level $M_i$, and the size of each module $m_j$ were recorded.

The single large module of the first level corresponds to the global network integration with the largest eigenvalue $\Lambda$. The second level with two modules supports local integration within each module and the segregation between them, which is weighted by a lower eigenvalue. With an increasing mode order, more modules reflect deeper levels of segregation, accompanied by smaller eigenvalues $\Lambda$. The integration and segregation component at each level can be defined as

$$H_i = \Lambda_i^2 M_i (1 - p_i)/N,$$

where $p_i = \frac{\sum_j |m_j - N/M_i|}{N}$ is a correction factor that takes into account heterogeneous modular sizes.

After this, the global integration component is taken from the first level $H_{in} = H_1/N$, and the segregation component is obtained as a sum from the 2nd to the $N$th levels, as $H_{se} = \sum_{i=2}^{N} H_i/N$. Finally, the local component of segregation $H_{se}^j$ and integration $H_{in}^j$ for each brain area $j$ can be obtained by weighting the integration components for the corresponding entries of the eigenvectors, as:

$$H_{in}^j = H_1 u_{1j}^2$$

$$H_{se}^j = \sum_{i=2}^{N} H_i u_{ij}^2,$$

where $u_{ij}$ is the $j$–th eigenvector entry at the $i$–th level. More details and previous applications of these measures can be found in [45,94].

## Statistical analyses

We used repeated measures correlation for assessing linear correlations between BF and LC fMRI time series and AAL90 ROIs, in all brain stages (W, N1, N2, and N3). This method is a statistical technique designed to assess the strength and direction of a linear relationship between two variables when data are collected repeatedly [35]. For group comparisons (e.g., W versus N1), we used paired T-tests. To minimize the likelihood of committing type I errors (false positives), we employed the Benjamini-Hochberg method for multiple comparisons correction. This correction was applied to paired comparisons and multiple correlations.

Given that statistical p-values could be artificially decreased by sample size in computer simulations (varying the number of seeds), instead of statistical tests, we report results using Cohen's D for effect size. Cohen's D is usually interpreted as representing a very small ( $< 0.2$ ), small ($0.2 < D < 0.5$), medium ($0.5 < D < 0.8$), large ($0.8 < D < 1.2$), very large ($1.2 < D < 2$) or huge ($2 < D$) effect size [95].

## Acknowledgments

F.L. is supported by Beca de Doctorado Nacional N° 21220708 (Agencia Nacional de Investigación y Desarrollo ANID, Chile). V.M. is supported by FONDECYT Exploración 13240170 and FONDECYT de Iniciación 11251578 (ANID, Chile). T.L. is supported by a German Research Foundation Grant (D.F.G.) Projektnummer 41466077, 526259959. J.K. received no specific funding for this work. S.O. is supported by Beca de Doctorado Nacional N°21241572 (ANID, Chile). D.B is supported by Beca de Doctorado Nacional N°21210914 (ANID, Chile). E.T is supported by FONDECYT Exploración 13240170 and FONDECYT Regular 1220995 (ANID, Chile). P.O. is supported by FONDECYT Regular 1241469 (ANID, Chile), FONDECYT Exploración 13240170, and the Advanced Center for Electrical and Electronic Engineering (AFB240002 ANID, Chile).

## Supporting information

**S1 Fig. Heatmaps of goodness of fit of all stages (columns), and all modalities (rows).** Variations are taken from the homogeneous optimal of W (G= 0.14, $\sigma$ = 7.7). Here, lower values (blue) indicate a better fit the empirical FC matrix of each state.
(TIFF)

**S2 Fig. Nodal integration (segregation) components from empirical and simulated data (simulated homogeneously, with map, and shuffled map).** Each point is a brain area, averaged across individuals (empirical) or seeds (simulated, N=50 seeds).
(TIFF)

## Author contributions

**Conceptualization:** Fernando Lehue, Carlos Coronel-Oliveros, Vicente Medel, Sebastián Orellana, Diego Becerra, Enzo Tagliazucchi, Patricio Orio.

**Data curation:** Fernando Lehue, Carlos Coronel-Oliveros, Enzo Tagliazucchi.

**Formal analysis:** Fernando Lehue, Carlos Coronel-Oliveros.

**Funding acquisition:** Patricio Orio.

**Investigation:** Fernando Lehue, Carlos Coronel-Oliveros, Thomas Liebe, Enzo Tagliazucchi, Patricio Orio.

**Methodology:** Fernando Lehue, Carlos Coronel-Oliveros, Vicente Medel, Sebastián Orellana, Enzo Tagliazucchi, Patricio Orio.

**Project administration:** Enzo Tagliazucchi, Patricio Orio.

**Resources:** Thomas Liebe, Jörn Kaufmann, Enzo Tagliazucchi, Patricio Orio.

**Software:** Fernando Lehue, Carlos Coronel-Oliveros, Enzo Tagliazucchi.

**Supervision:** Enzo Tagliazucchi, Patricio Orio.

**Validation:** Fernando Lehue, Carlos Coronel-Oliveros, Patricio Orio.

**Visualization:** Fernando Lehue, Vicente Medel.

**Writing – original draft:** Fernando Lehue, Diego Becerra.

**Writing – review & editing:** Fernando Lehue, Carlos Coronel-Oliveros, Vicente Medel, Thomas Liebe, Jörn Kaufmann, Sebastián Orellana, Diego Becerra, Enzo Tagliazucchi, Patricio Orio.

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
