## [Decision Letter · Decision Letter 0]

6 May 2025

PCOMPBIOL-D-25-00223

Noradrenaline and Acetylcholine shape Functional Connectivity organization of NREM substages: an empirical and simulation study

PLOS Computational Biology

Dear Dr. Orio,

Thank you for submitting your manuscript to PLOS Computational Biology. After careful consideration, we feel that it has merit but does not fully meet PLOS Computational Biology's publication criteria as it currently stands. Therefore, we invite you to submit a revised version of the manuscript that addresses the points raised during the review process.

Please submit your revised manuscript within 60 days Jul 06 2025 11:59PM. If you will need more time than this to complete your revisions, please reply to this message or contact the journal office at ploscompbiol@plos.org. Please include the following items when submitting your revised manuscript:

We look forward to receiving your revised manuscript.

Kind regards,

Maxim Bazhenov

Academic Editor

PLOS Computational Biology

Hugues Berry

Section Editor

PLOS Computational Biology

**Journal Requirements:**

At this stage, the following Authors/Authors require contributions: Vicente Medel. Please ensure that the full contributions of each author are acknowledged in the "Add/Edit/Remove Authors" section of our submission form.

5) We have noticed that you have uploaded Supporting Information files, but you have not included a list of legends. Please add a full list of legends for your Supporting Information files after the references list.

Potential Copyright Issues:

i) Figures 1C, and 2. Please confirm whether you drew the images / clip-art within the figure panels by hand. If you did not draw the images, please provide (a) a link to the source of the images or icons and their license / terms of use; or (b) written permission from the copyright holder to publish the images or icons under our CC BY 4.0 license. Alternatively, you may replace the images with open source alternatives. See these open source resources you may use to replace images / clip-art:

7) In the online submission form, you indicated that "fMRI data has been previously published in Tagliazucchi et al., 2012 (10.1016/j.neuroimage.2012.06.036).Structural connectome data was kindly provided by Dr. Gustavo Deco (Deco et al., 2018, 10.1016/j.cub.2018.07.083) and is available upon request." All PLOS journals now require all data underlying the findings described in their manuscript to be freely available to other researchers, either

1. In a public repository

2. Within the manuscript itself

3. Uploaded as supplementary information.

8) Please amend your detailed Financial Disclosure statement. This is published with the article. It must therefore be completed in full sentences and contain the exact wording you wish to be published.

1) State what role the funders took in the study. If the funders had no role in your study, please state: "The funders had no role in study design, data collection and analysis, decision to publish, or preparation of the manuscript.".

9) Please ensure that the funders and grant numbers match between the Financial Disclosure field and the Funding Information tab in your submission form. Note that the funders must be provided in the same order in both places as well. Currently, " C.C. is supported by BrainLat/NIH and GBHI funding" is missing from the Funding Information tab.

**Reviewers' comments:**

Reviewer's Responses to Questions

Reviewer #1: This study explores how the neuromodulators Acetylcholine (ACh) and Noradrenaline (NA) influence brain connectivity during NREM sleep stages using a whole-brain model. The authors present a novel approach by integrating ACh and NA into a model informed by an empirical brain connectome and neurochemical maps. They identify key changes in functional connectivity (FC), such as decreased LC connectivity with the cortex in the N2 and N3 stages, and reduced BF connectivity with the cortex in N3. The study also reports a transition to a more integrated FC in N1 and a more segregated state in N3 compared to wakefulness. The model adjustments underscore the importance of region-specific neurotransmitter effects on brain connectivity.

The manuscript is generally well-written. I have made several comments listed below.

Major

Is it possible to model other key neurotransmitters in relation to BOLD fluctuations? For example, GABA, which increases after NREM sleep compared to wakefulness and changes in the opposite direction of ACh and NA.

To ensure that the masks for BF and LC are appropriately chosen and positioned, they should be displayed on the BOLD data of a representative participant.

Lines 194-196: How did the authors decide on the parameters for δG and δσ? In addition, δG for N2 in the text is different from that shown in Fig. 4. In the heterogeneous modulation analyses, it would be useful to show the δG and δσ maps across the brain.

Fig. 5C: Is it possible to compare the similarity among states for the same modality, e.g., the map strategy? It seems that the correlations between simulated and empirical integration (segregation) in N2 are the lowest. I would appreciate it if the authors could comment on this. Similar concerns raises for Fig. 3C.

Minor

The full spelling of ACh and NA should be provided at their first appearance, i.e., line 22.

Reviewer #2: The paper of Lehue et al. investigates the role of ACh/NA modulation in shaping FC across different sleep stages. The authors leverage experimental fMRI data to tune whole brain mean-field model in order to explain FC changes by ACh/NA release and present model covering all major sleep stages in humans. They show interesting point that using heterogenous map of neuromodulatory release based on presumed LC/BF connectivity improves the fit to experimental data, thus suggesting the role of neuromodulation in shaping sleep FC dynamics. They also connect their story with a broader thread of studies showing ACh/NE influences on segregation/integration properties of the network.

I find that the paper is timely and on an important topic and although I have a major concern to what extent can be their manipulation called "ACh/NA modulation" I believe this paper is of interest.

These are my points with decreasing order of importance:

1) I have major concern to what extent are the changes of global coupling and slope of input-output function of excitatory population realistic proxy for the ACh/NA modulation, because the effects of both ACh/NA are very complex. E.g. authors defend NA modulation by referring to decades old slope of input-output function proposal, but lot have been shown since then - there is a maze of opposing (enhances/reduces syn. transmission) effects based on alpha vs beta receptors, region identity (PFC vs rest), layer sensitivity, etc which won't be covered by simply estimating map/connectivity from LC; the case of ACh won't be less complex. I am happy to be proved wrong, but covering all this looks beyond the scope of this work, so it seems that the casual reader should be warned in the beginning of the paper what "ACh/NA" effects actually translate to and that certain critical stance is warranted, instead of unsupported claims that ignoring specific receptors effects does not change general effect of tonic levels of ACh/NA (l.270).

2) There is a certain tension in rodent literature between traditional view that NA is reduced during NREM and more recent experimental work correcting this at several fronts -

a) NA is indeed somewhat weaker in neocortical brain areas, but still overlaps with values at wake,

b) for some regions (sensory thalamus) NA levels are even higher in NREM than in quiet wakefulness

c) NA level is not tonic, but dynamic

To what degree presented results hold, if a) NA reduction in NREM is very slight b) oscillates in infraslow regime?

I see the comment in manuscript (l. 442) that LC->thalamic connectivity needed artificial decrease of strength. Is there some possible connection with point b)?

3) Data availability issues:

a) Authors claim fMRI dataset is available in Tagliazucchi et al., 2012. I could find the paper describing results, but no data. Is the availability statement misleading or can the authors provide better pointers where the experimental data are deposited?

b) It looks to me that in the current form the simulation code at github is not reproducible and important private modules/files are missing (e.g. BOLDModel, "integration_segregation/", "RSN_AAL_Enzo.txt", etc). If they are not private, access to them should be described in dependencies of the code.

c) The connectome itself looks present after some digging, but should be better exposed. It would be nice if the files with a) global connectome b) maps/connectivity from LC/BF to the cortical sites are explicitly described/offered - either in methods or in github readme. Especially not so common explicit LC/BF -> cortex connectivity/maps in AAL could be useful for the modelling community.

Minor points:

- Do authors have a sense how much is using both hemispheres fundamental for the present results (i.e. if instead of artificially amending homotopic connections between hemispheres, only single hemisphere was used)?

- I could not find explicit description how were the structural connections from LC to the rest of AAL areas established in methods (though I found relevant reference later elsewhere in the manuscript). Glancing over Liebe et al. paper suggests they do not seem to provide it in the right template(?).

- The whole discussion section "Sleep as a model of consciousness" reads somewhat superfluous and I had difficulty to connect it with the actual results of the paper.

- The numbering of both figures/their panels does not fit the order they are introduced in the text and forces the reader to do chaotic jumps (e.g. 1B,1A,1C,5B,5A,2,3B,3,4,3C,5C,...)

- Probably missing something obvious, but why are signs of +-ACh/-+NA opposite in Fig 3B, opposite y-axes?

- Fig 3A - stage labels on top of the dots would be helpful.

- Caption of panel Fig3A contains description of violin plots, which should be probably elsewhere.

Typos:

l. 12 W -> (W)

l. 17 orchastrate

l. 159 - Figure -> Fig

l. 177 "the the"

l. 242 ")"

l. 347 N3, N1

l. 389 mm^3

l. 440 "("

l. 441 "document" sounds strange

**Have the authors made all data and (if applicable) computational code underlying the findings in their manuscript fully available?**

Reviewer #1: Yes

Reviewer #2: **No: **I put detailed comments in my response. To sum up some modelling things seem missing, but looks easily fixable. Experimental material is missing altogether but is not central to this paper and might not be even permitted for publishing.

PLOS authors have the option to publish the peer review history of their article (what does this mean?). If published, this will include your full peer review and any attached files.

Reviewer #1: **Yes: **Qihong Zou

Reviewer #2: No

**Figure resubmission:**
---

## [Decision Letter · Decision Letter 1]

18 Sep 2025

PCOMPBIOL-D-25-00223R1

Noradrenaline and Acetylcholine shape Functional Connectivity organization of NREM substages: an empirical and simulation study

PLOS Computational Biology

Dear Dr. Orio,

Thank you for submitting your manuscript to PLOS Computational Biology. After careful consideration, we feel that it has merit but does not fully meet PLOS Computational Biology's publication criteria as it currently stands. Therefore, we invite you to submit a revised version of the manuscript that addresses the points raised during the review process.

Reviewer#2 judges that their concern about data and code availability has not been addressed (the git repository seems not to have been restructured and the Zenodo repository has not been made accessible to them). Please proceed to the necessary changes.

Please submit your revised manuscript within 30 days Nov 18 2025 11:59PM. If you will need more time than this to complete your revisions, please reply to this message or contact the journal office at ploscompbiol@plos.org. Please include the following items when submitting your revised manuscript:

We look forward to receiving your revised manuscript.

Kind regards,

Hugues Berry

Section Editor

PLOS Computational Biology

Hugues Berry

Section Editor

PLOS Computational Biology

**Reviewers' comments:**

Reviewer's Responses to Questions

Reviewer #1: All comments were thoroughly addressed.

Reviewer #2: I see the authors response to the major point #3 (Data availability) from the first review as inadequate, in particular:

1) Authors claim that git repository was restructured. I do not see any changes/commits in github repository so points 3b & 3c were not in fact addressed. Perhaps the authors forget to push the changes to the public repo?

2) New Zenodo deposit addressing 3a has restricted access (which is fine in the pre-accepted stage), but the available description of the dataset looks very insufficient for anyone to make sense of the deposited data. Given the repeated issues with data deposits for this paper I would appreciate Zenodo's hidden link for reviewers to review the deposit description details to decide whether they give the community chance to actually validate the paper claims.

**Have the authors made all data and (if applicable) computational code underlying the findings in their manuscript fully available?**

Reviewer #1: Yes

Reviewer #2: **No: **As written in the response to the authors, my concerns relating with the code/data availability were not yet addressed, but given the positive nature of their first response it seems more as an unintended omission which can be easily fixed.

PLOS authors have the option to publish the peer review history of their article (what does this mean?). If published, this will include your full peer review and any attached files.

Reviewer #1: **Yes: **Qihong Zou

Reviewer #2: No

**Figure resubmission:**
---

## [Editor Report · Decision Letter 2]

14 Oct 2025

Dear Prof. Orio,

We are pleased to inform you that your manuscript 'Noradrenaline and Acetylcholine shape Functional Connectivity organization of NREM substages: an empirical and simulation study' has been provisionally accepted for publication in PLOS Computational Biology.

Best regards,

Hugues Berry

Section Editor

PLOS Computational Biology

Hugues Berry

Section Editor

PLOS Computational Biology

---

## [Editor Report · Acceptance letter]

PCOMPBIOL-D-25-00223R2

Noradrenaline and Acetylcholine shape Functional Connectivity organization of NREM substages: an empirical and simulation study

Dear Dr Orio,

I am pleased to inform you that your manuscript has been formally accepted for publication in PLOS Computational Biology. Your manuscript is now with our production department and you will be notified of the publication date in due course.

With kind regards,

Zsofia Freund
